# The combined DNA and RNA synthetic capabilities of archaeal DNA primase facilitate primer hand-off to the replicative DNA polymerase

Mark D. Greci 🄳 [1], Joseph D. Dooher[2] & Stephen D. Bell 🄳 [1,3✉]

Replicative DNA polymerases cannot initiate DNA synthesis de novo and rely on dedicated RNA polymerases, primases, to generate a short primer. This primer is then extended by the DNA polymerase. In diverse archaeal species, the primase has long been known to have the ability to synthesize both RNA and DNA. However, the relevance of these dual nucleic acid synthetic modes for productive primer synthesis has remained enigmatic. In the current work, we reveal that the ability of primase to polymerize DNA serves dual roles in promoting the hand-off of the primer to the replicative DNA polymerase holoenzyme. First, it creates a 5′-RNA-DNA-3′ hybrid primer which serves as an optimal substrate for elongation by the replicative DNA polymerase. Second, it promotes primer release by primase. Furthermore, modeling and experimental data indicate that primase incorporates a deoxyribonucleotide stochastically during elongation and that this switches the primase into a dedicated DNA synthetic mode polymerase.

---

[1] Department of Biology, Indiana University, Simon Hall MSB, 212S Hawthorne Drive, Bloomington, IN, 47405, USA. [2] Independent Scholar, Indiana University, Simon Hall MSB, 212S Hawthorne Drive, Bloomington, IN, 47405, USA. [3] Department of Molecular and Cellular Biochemistry, Indiana University, Simon Hall MSB, 212S Hawthorne Drive, Bloomington, IN, 47405, USA. ✉email: stedbell@indiana.edu

The inability of replicative DNA polymerases to initiate DNA synthesis de novo means that cellular DNA replication systems require dedicated RNA polymerases, primases, to generate a short oligoribonucleotide primer that is then transferred to a DNA polymerase. In bacteria, the TOPRIM-fold-containing DNA primase, DnaG, transfers the primer to a C-family replicative DNA polymerase holoenzyme via a "three-point switch" involving the action of the single strand binding protein as an intermediary[1]. The primases of archaea and eukaryotes are related to one another and are entirely distinct from the bacterial DnaG-type primases[2]. The Archaeal/eukaryotic primases (AEPs) belong to an extended family of nucleic acid polymerases that are encoded in cellular and extra-chromosomal element genomes[3,4]. One broadly conserved AEP family member, PrimPol, has been demonstrated to act as a DNA polymerase in replication restart and lesion bypass in human cells[5].

The canonical eukaryotic primase possesses two subunits; PriS which contains the polymerase active site, and PriL which contains a helical bundle domain (HBD) that is required for initiation of primer synthesis but which is dispensable for RNA elongation by primase[6,7]. PriS and PriL are found in the context of the pol α complex, in the presence of the B-family DNA polymerase α and its B-subunit[8]. An internal hand-off within this tetrameric complex leads to the transfer of the RNA primer between primase and DNA pol α. The latter enzyme extends the short RNA primer with DNA, prior to transfer of the hybrid nucleic acid to the replicative DNA polymerase. Notably, DNA Pol α exhibits a strong preference for extending a RNA primer on a DNA template. The thumb domain of Pol α plays a key role in recognizing and stabilizing the interaction with the A-form RNA-DNA heteroduplex and destabilizing the interaction once the polymerase reaches the B-form DNA that it has just synthesized[9], thereby providing an elegant mechanism for simultaneously constraining the length of DNA synthesized by Pol α and facilitating hand-off to the replicative DNA polymerase.

In *Saccharolobus solfataricus* (formerly *Sulfolobus solfataricus*) —a member of the crenarchaeal order *Sulfolobales*, PriS and PriL form a complex with a third small protein, PriX[10,11]. Significantly, PriX is a structural ortholog of the HBD of the large subunit of eukaryotic DNA primase. PriX has been demonstrated to bind the initiating nucleotide via its 5′ triphosphate groups, allowing de novo initiation of nucleic acid synthesis[10]. Additionally, PriX retains a grip on the 5′ end of the elongating primer, providing a caliper-like mechanism for constraining primer length[12]. How the primer is transferred to the B-family replicative DNA polymerase, PolB1-HE, is currently unknown. Interestingly, the primases of a number of archaea have been demonstrated to be relatively promiscuous with regard to their ability to initiate and elongate either RNA or DNA[11,13–15]. This observation has led to the suggestion that archaeal primase could have roles beyond priming, and a recent study in a reconstituted system with proteins derived from the anaerobic *Pyrococcus abyssi* has indicated that PriSL plays a role in the bypass of oxidative damage, in the form of 8-oxo-dG[15]. It has also been proposed that the archaeal primase, via its dual synthetic capabilities, could switch between RNA and DNA synthetic modes to generate a hybrid primer[11–13].

DNA polymerases in archaea have a complex phyletic distribution, with most archaea, other than the phylum crenarchaea, possessing a D-family polymerase that is the sole polymerase essential for viability in the species *Thermococcus kodakarensis* and *Methanococcus maripaludis* and thus likely the replicative enzyme[16,17]. The crenarchaea lack PolD but typically possess at least two B-family DNA polymerases. Species in the order *Sulfolobales*, members of the crenarchaea, encodes three B-family polymerases, PolB1, PolB2 and PolB3. Genetic experiments have shown that only PolB1 is essential for viability, with PolB2 and PolB3 playing roles in DNA repair processes[18–20]. In the *Sulfolobales*, PolB1 forms a stable complex with two small sub-units, PBP1 and PBP2, forming a heterotrimeric holoenzyme, PolB1-HE[21].

In the current work, we reconstitute primer hand-off between primase and polymerase using purified recombinant *Saccharolobus solfataricus* PriSLX and the replicative B-family DNA polymerase holoenzyme, PolB1-HE. Our data reveal a model for the primer hand-off and a functional explanation for the hitherto enigmatic DNA synthetic capability of archaeal primase.

More specifically, we reveal that PolB1-HE is inefficient at elongating a RNA primer with DNA. However, the inclusion of PriSLX robustly stimulates DNA synthesis. We demonstrate that 5′-RNA-DNA-3′ hybrid primer is readily elongated by PolB1-HE and that this hybrid molecule can be generated by primase. Notably, dNMP incorporation by PriSLX promotes disengagement of PriSLX from the primer, thus facilitating primer release and hand-off to PolB1-HE.

## Results

### De novo DNA synthesis is dependent on RNA synthesis by PriSLX.
Using purified recombinant PriSLX and DNA polymerase holoenzyme, PolB1-HE, we could reconstitute primase-dependent DNA synthesis in vitro (Fig. 1a). This reconstituted system was dependent on the presence of catalytically active PriSLX and PolB1-HE (Fig. 1a—lane 7), as the inclusion of derivatives of either primase or polymerase possessing substitutions of alanine for catalytic site aspartates abrogated the reaction. Next, we determined whether the reconstituted system was dependent on RNA synthesis by the primase. As demonstrated in Fig. 1b, the presence of both NTPs and dNTPs led to an $8.00 \pm 0.04$-fold greater yield of DNA product than when NTPs were omitted from the reaction (Fig. 1b—lane 2 compared to lane 3). We note that the catalytic activities of both enzymes are required, as inclusion of, "polymerase dead" (PD) versions of the primase and polymerase that have amino acid substitutions in the active site aspartate residues, D101,103A and D655,657A, respectively, abolish product yield.

### DNA synthesis by primase facilitates hand-off of a RNA primer to DNA polymerase.
Given that RNA synthesis by primase is required for maximal activity in our reconstituted system, we next tested the ability of PolB1-HE to extend pre-formed RNA or DNA primers in reactions containing dNTPs, in the absence of additional NTPs. Our previous work has demonstrated that primase synthesizes primers between roughly 10 and 20 nucleotides in length[10]. Accordingly, we tested whether 5′-fluorescently-labeled synthetic RNA or DNA oligonucleotides of 19 nucleotides in length, representing mature or near-mature-length primer, could support extension by PolB1-HE (Fig. 2a—lanes 1–6). In agreement with previous reports[21,22], PolB1-HE could extend a DNA primer (Fig. 2a—lanes 4–6). However, PolB1-HE was inefficient at extending a purely RNA primer (Fig. 2a—lanes 1–3). Strikingly, when PriSLX was added to the reaction, PolB1-HE and PriSLX mediated robust extension of the RNA primer (Fig. 2a—lanes 7–9). The ability of PriSLX to stimulate DNA polymerization in the absence of ongoing ribonucleotide incorporation led us to speculate that the DNA synthetic capability of primase could be responsible for this enhanced yield of full-length primer extension products.

To test whether DNA—3′ tailing of an otherwise RNA primer might influence extension of a primer by PolB1-HE, we next analyzed the ability of PolB1-HE to elongate 5′-fluorescently-labeled synthetic 19 nt hybrid primers with sequences analogous

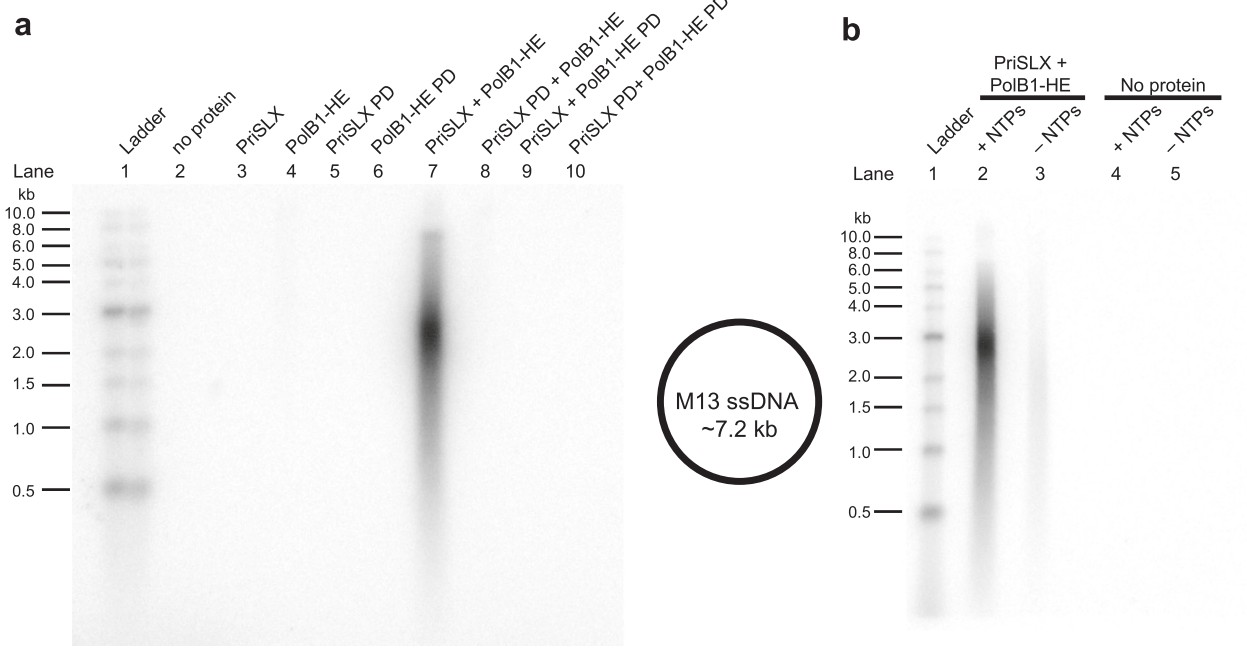

**Fig. 1 In vitro replication by PriSLX and PolB1-HE. a** Left panel: replication reactions containing 200 μM NTPs; 30 μM dNTPs; 2 μCi/10 μL α-$^{32}$P-dATP; 25 nM M13 ssDNA template; and 25 nM of protein(s) as indicated. 30 min at 75 °C. Right panel: representation of M13 ssDNA template. The suffix "PD", Polymerase Dead, indicates versions of the primase and polymerase that have amino acid substitutions in the active site aspartate residues, D101,103A and D655,657A, respectively, that render them inactive. Two independent replicates of this experiment were performed. **b** Replication reactions with or without 200 μM NTPs as indicated; 30 μM dNTPs; 2 μCi/10 μL α-$^{32}$P-dATP; 25 nM M13 ssDNA template; and 25 nM of protein(s) as indicated. 30 min at 75 °C. Three independent replicates of this experiment were performed.

to the 19 nt RNA- or DNA primer used in Fig. 2a—lanes 1–16. The hybrid primers were mostly contiguous RNA but had a short 2 or 1 dNMP 3′-tail: termed "17r/2d nt" and "18r/1d nt", respectively. While PolB1-HE alone did not give the same yield of full-length product from hybrid primers when compared with DNA primer (Fig. 2a—lanes 17–22 compared to Fig. 2a—lanes 4–6), it nevertheless was able to extend the hybrid primers appreciably more than the RNA primer (Fig. 2a—lanes 17–22 compared to lanes 1–3). We observe a slight preference for extension of 17r/2d over 18r/1d nt hybrid primer (Fig. 2a—lane 18 compared to 21). In contrast to PriSLX addition dramatically stimulating RNA primer elongation, addition of PriSLX to the hybrid primer reactions only minimally further stimulated hybrid primer elongation (Fig. 2b—lanes 23–28 compared to lanes 17–22).

In light of the ability of PolB1-HE to elongate DNA or hybrid primers terminating with a 3′-DNA tail, we next considered how PriSLX might facilitate stimulation of RNA primer elongation. One possibility is that PriSLX synthesizes a DNA tail, forming an optimal substrate for extension by PolB1-HE. Alternatively, PriSLX could simply bind to the primer-template construct and facilitate recruitment of PolB1-HE.

To discriminate between these two, not necessarily mutually exclusive, possibilities, we performed reactions containing a PriSLX derivative, PriSLX PD, in which the catalytic site contained alanine substitutions at the catalytic residues, Asp 101 and Asp 103. The PriSLX PD is catalytically inactive but retains primer-template binding affinity in the ~100 nanomolar range (Supplementary Fig. 1a) comparable to wild-type PriSLX (Fig. 5a). If PriSLX stimulates RNA primer elongation solely by a recruitment mechanism, we would anticipate that PriSLX PD would stimulate primer extension by PolB1-HE.

As revealed in Fig. 2b, while catalytic PriSLX addition to the RNA primer reactions elevates the yield of primer extension

products (Fig. 2b, 4–6), addition of PriSLX PD did not (Fig. 2b—lanes 7–9). Thus, PriSLX is not acting by simply facilitating recruitment of PolB1-HE to the primer-template junction. In addition, we performed fluorescence polarization measurements of the DNA polymerase holoenzyme binding to the various primer-template substrates. This revealed that PolB1-HE was able to bind all four substrates with 60 to 140 nM binding affinity, suggesting that substrate binding dynamics does not govern the observed preference for DNA or DNA-terminating primers. Interestingly, the strongest affinities were observed for the primer-templates with RNA-primer or -DNA-terminating 17r/2d hybrid-primer.

Intriguingly, we noted that PriSLX PD addition to the hybrid-primed reactions did give a very modest enhancement of polymerization product beyond that seen with PolB1-HE alone (Fig. 2c—lanes 7–12 compared to 1–6). Given that wild-type PriSLX also minimally stimulated hybrid primer elongation, we further tested if the slight stimulation was due to bona fide catalytic or structural function of PriSLX or was, alternatively, a non-specific effect. As demonstrated in Supplementary Fig. 2, which utilizes BSA as a non-specific control, the addition of either PriSLX or PriSLX PD to hybrid-primer reactions does not seem to stimulate elongation to any greater level than that observed with an equal molar concentration of BSA (Supplementary Fig. 2—lanes 1–12 and 17–28).

Additionally, in hybrid primer elongation reactions proceeding with sub-saturating levels of PolB1-HE and a molar-excess of PriSLX or PriSLX PD, only catalytically active PriSLX detectably stimulates elongation (Fig. 3—lanes 37–44 and lanes 53–60). Thus, PriSLX does not simply stimulate the reaction by recruiting PolB1-HE to the hybrid primer/template junction. Consistent with a lack of a recruitment role, DNA primer elongation reactions are actually inhibited by a high excess of PriSLX or PriSLX PD (33-fold over PolB1-HE) (Fig. 3—lanes 21 and 25).

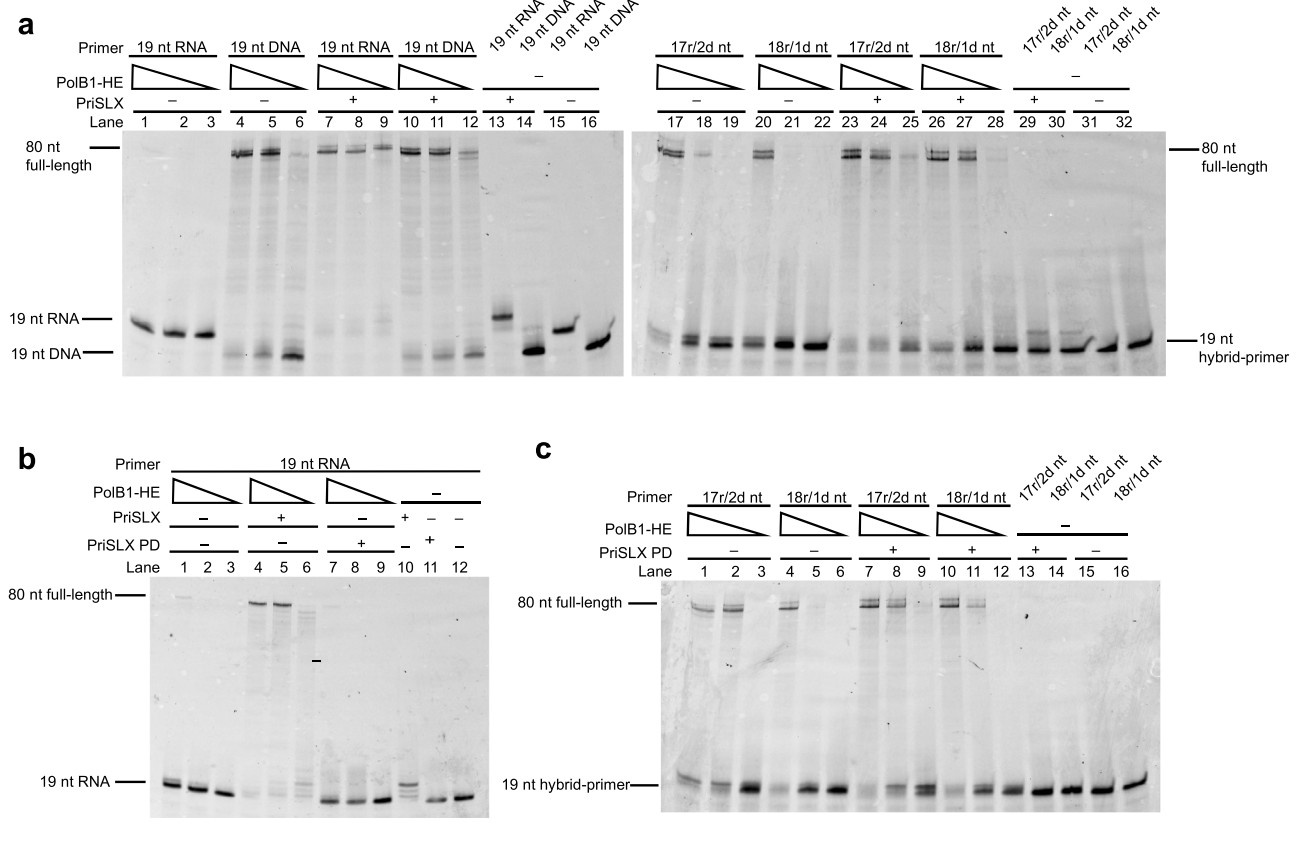

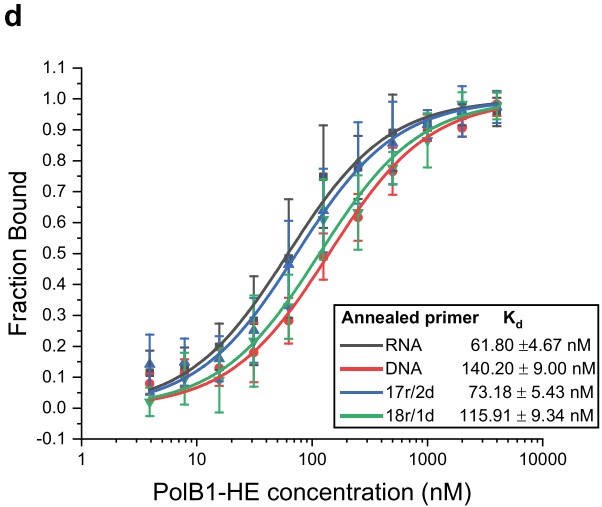

**Fig. 2 DNA-synthetic mode PriSLX stimulates RNA-primer extension. a–c** Primer elongation reactions with 100 µM dNTPs; 25 nM 5′-cy5-labeled primer as indicated annealed to 80 nt template; PolB1-HE titration (25, 12.5, 6.25 nM) as indicated; 25 nM PriSLX as indicated; and PriSLX PD (Polymerase Dead) as indicated. Reactions were incubated for 5 min at 75 °C prior to termination and electrophoresis on denaturing polyacrylamide gels. Three independent replicates of this experiment were performed. **d** $K_d$ determination by fluorescence polarization. Titration series of PolB1-HE as indicated with indicated 5′-Cy5-labeled primer annealed to 80 nt template. Data points indicate the mean value and error bars represent standard deviation. Four independent replicates of this experiment were performed.

As the hybrid primer elongation reactions proceeding with subsaturated PolB1-HE are stimulated by molar-excess PriSLX addition (Fig. 3—lanes 37–40 and 53–56) but PriSLX does not recruit PolB1-HE to the hybrid primer substrate, DNA synthesis by PriSLX stimulates hybrid primer elongation by PolB1-HE. Thus, PriSLX may synthesize 3′-DNA tails longer than the 1 or 2 terminating dNMPs found in our preformed hybrid primers, providing a better target for recognition by PolB1-HE.

Taken together, our data support a model in which PriSLX extends a RNA primer with a 3′-DNA tail, thus generating a 5′-RNA-DNA-3′ hybrid primer optimal for enacting productive hand-off and elongation.

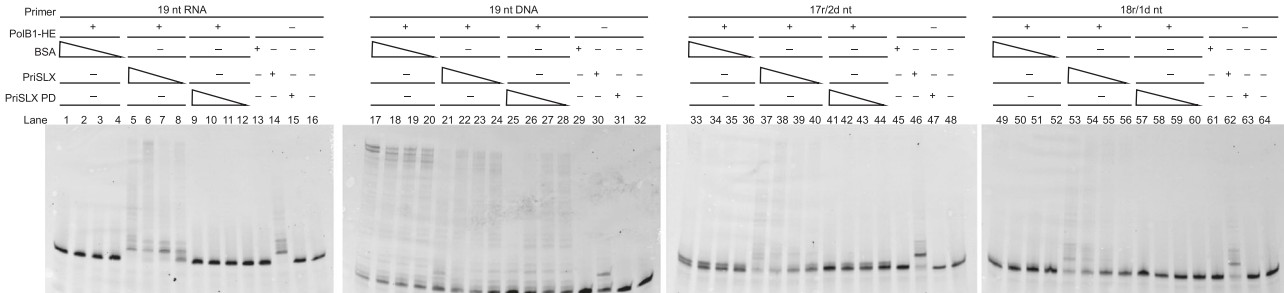

**Fig. 3 Primer elongation by sub-saturating amounts of PolB1-HE.** Primer elongation reactions of 100 μM dNTPs; 25 nM 5′-cy5-labeled primer as indicated annealed to 80 nt template; 6 nM PolB1-HE as indicated; BSA titration (200, 100, 50, 25 nM) and control (200 nM) as indicated; PriSLX titration (200, 100, 50, 25 nM) and control (200 nM) as indicated; and PriSLX PD (Polymerase Dead) titration (200, 100, 50, 25 nM) and control (200 nM) as indicated. Reactions were incubated for 5 min at 75 °C prior to termination and electrophoresis on denaturing polyacrylamide gels. Three independent replicates of this experiment were performed.

**DNA polymerization by primase**. The above data demonstrate that both the RNA and DNA synthetic capabilities of PriSLX are functionally important. We, therefore, wished to test the possibility that the full-length extension products that we observe could arise from primer extension by PriSLX alone. However, as seen in Fig. 2b (lane 10), omission of PolB1-HE leads to DNA extension products of up to 3 nt by PriSLX. Given that it was formally possible that PolB1-HE could stimulate the polymerization activity of PriSLX, we tested the effect of a polymerase-inactive version of PolB1-HE, PolB1-HE PD (in which the catalytic residues Asp655 and Asp657 are substituted with alanine). While inclusion of wild-type PolB1-HE in assays with PriSLX led to full-length elongation (Fig. 2), addition of inactive PolB1-HE PD to RNA primer reactions with PriSLX did not yield any elongation products beyond the initial +3 dNMPs seen with PriSLX alone (Fig. 4a— lanes 4–5 compared to 2–3). The altered profile of PriSLX extension products, with a relative reduction in the yield of the +3 product, is likely due to the exonuclease activity of PolB1-HE PD (see also Fig. 4a—lanes 12 and 15). We also note that PolB1-HE PD addition does not stimulate DNA- or hybrid primer elongation by PriSLX beyond the low basal level (Fig. 4a—12–13; 20–21; 28–29). Thus, PolB1-HE does not appear to directly influence the nucleic acid polymerization activity of PriSLX.

We next examined PriSLX polymerase activity over an extended 20-min time-course (Fig. 4b). Our previous work has revealed that PriSLX grips the 5′-triphosphate of a primer, thus constraining primer length as the elongation site translocates from the initiation site and limiting maximum product size to ~20 nucleotides[12]. Remarkably, even though primers in our current assays possessed a 5′-florescent label, and thus are presumably not recognized by PriX, an overwhelming majority of 19 nt RNA primer elongation remained at +3 nucleotides (22 nt product) for both RNA and DNA synthetic modes (Fig. 4b—lanes 10 and 20), suggesting that structural aspects of length constraint remain engaged despite the fluorescent label. Halting of synthesis is especially pronounced in the DNA synthetic mode time course, where the +3 nucleotide product accumulates rapidly and is minimally converted to greater length product at extended time points (Fig. 4b—lanes 18–20). Similarly, the accumulation and persistence of the +3 nucleotide product is evident in the RNA synthetic mode time course, as well (Fig. 4b—lanes 8–10). This product length constraint confirms that, as we inferred when designing the primers, 19 nt length primer corresponds to near-maximal length primers.

Moreover, the time course underscores the preference of PriSLX to act as a DNA rather than a RNA polymerase—at least as the RNA primer length approaches the end of the length

constraints. Two minutes into the DNA-synthetic mode time course, nearly all the original RNA primer has been converted to DNA tailed, hybrid-chain product (Fig. 4b—lane 16). In contrast, 20 min into the RNA-synthetic mode time course, the most abundant species is still unutilized RNA primer (Fig. 4b—lane 10). Taken together, this demonstrates that in the extension of a 19 nt RNA primer, PriSLX has greater DNA polymerase activity.

To further characterize PriSLX synthetic capabilities, we analyzed the ability of PriSLX to act as an RNA or DNA polymerase to elongate the 19 nt RNA-, DNA-, and hybrid primers (Fig. 4c). Strikingly, PriSLX—acting as an RNA polymerase—only appreciably elongates the RNA primer (Fig. 4c—lanes 1–4 compared to 5–16) whereas, acting as a DNA polymerase, PriSLX elongates the RNA primer (Fig. 4c—lanes 17–20) and, to a lesser extent, the DNA- and hybrid primers (Fig. 4c—21–24; 25–28; 29–32). Thus, when the 3′-terminal nucleotide in the primer is dNMP, PriSLX favors elongation in DNA synthetic-mode. We therefore infer that once a dNMP is incorporated, PriSLX commits to DNA-synthetic mode to synthesize an uninterrupted 3′-DNA tail.

As demonstrated by the previous time course (Fig. 4b) and the current assay (Fig. 4c), PriSLX is a more active DNA polymerase than RNA polymerase in the single nucleotide elongation of a 19 nt RNA primer (Fig. 4c—lanes 17–20 compared to lanes 1–4)—at least when NTP and dNTP concentrations are equivalent (see below). Therefore, a switch to DNA-synthetic mode downstream of the initiating NTP could be intrinsic to primer synthesis.

Furthermore, while PriSLX readily elongates RNA, DNA, and hybrid-primers acting as a DNA polymerase, the yield of product arising from extension of the DNA primer and the DNA-terminating hybrid primers is appreciably less than the extension of the RNA primer (Fig. 4c—lanes 21–24; 25–28; 29–32 compared to 17–20). Given the apparent commitment to DNA-synthetic mode and the reduced elongation of a primer with 3′ dNMP(s), we hypothesized the existence of a dNMP-incorporation-driven negative feedback loop in which dNMP incorporation commits PriSLX to DNA tail synthesis and subsequent synthesis termination, thus facilitating primer hand-off to PolB1-HE.

**dNMP incorporation by PriSLX drives primer disengagement and hand-off**. To test the hypothesis of the existence of a dNMP-incorporation-driven negative feedback loop and its potential influence on hand-off, we first measured the affinity of PriSLX for primer-template substrates containing the 19 nt RNA-, DNA-, and hybrid-primers. Conceivably, negative feedback driven by dNMP incorporation could proceed via a simple binding

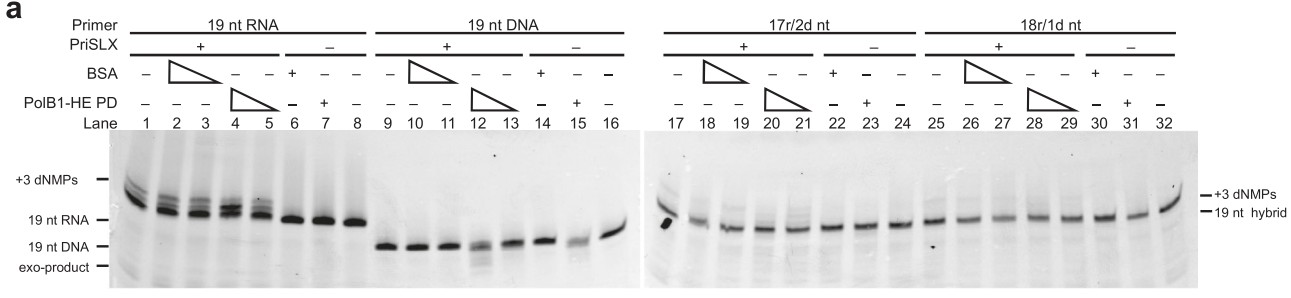

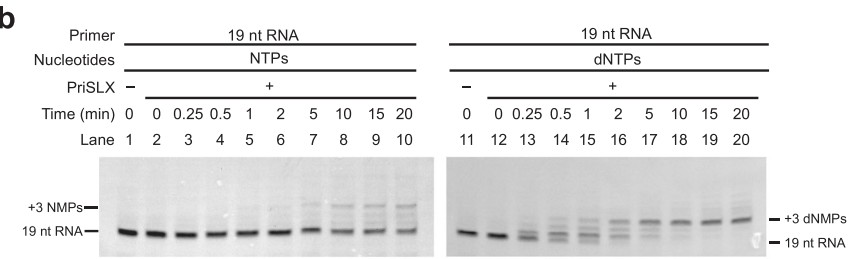

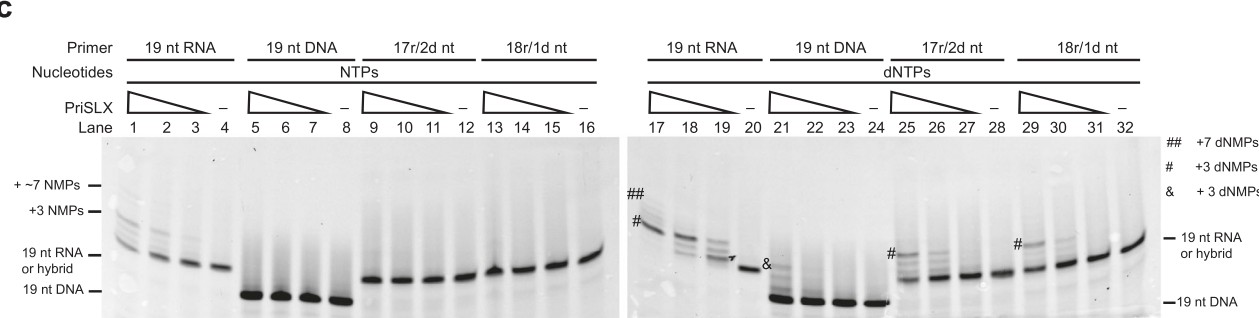

**Fig. 4 PriSLX elongation activity. a** Primer elongation reactions of 100 μM dNTPs; 25 nM 5′-cy5-labeled primer as indicated annealed to 80 nt template; 25 nM PriSLX as indicated; BSA titration (25, 12.5 nM) and control (25 nM) as indicated; and PolB1-HE PD (Polymerase Dead) titration (25, 12.5 nM) and control (25 nM) as indicated. Reactions were incubated for 5 min at 75 °C prior to termination and electrophoresis on denaturing polyacrylamide gels. Three independent replicates of this experiment were performed. **b** Primer elongation reactions of 100 μM NTPs or dNTPs as indicated; 25 nM 5′-cy5-labeled primer annealed to 80 nt template; and 25 nM PriSLX. Reaction times as indicated at 75 °C prior to termination and electrophoresis on denaturing polyacrylamide gels. Three independent replicates of this experiment were performed. **c** Primer elongation reactions of 100 μM NTPs or dNTPs as indicated; 25 nM 5′-cy5-labeled primer as indicated annealed to 80 nt template; and PriSLX titration (50, 25, 12.5 nM) as indicated. Reactions were incubated for 5 min at 75 °C prior to termination and electrophoresis on denaturing polyacrylamide gels. Four independent replicates of this experiment were performed.

mechanism: PriSLX could have lower affinity for dNMP-terminating substrate, and thus may not bind as tightly to the primer-template junction. However, when we tested PriSLX binding affinity, it displayed similar binding to all the substrates, with binding affinity ranging from 80 to 140 nM (Fig. 5a), thereby eliminating binding dynamics as a driver of the dNMP-driven negative feedback.

Next, we performed steady-state kinetic analyses of PriSLX in RNA and DNA synthetic mode (Fig. 5b) to examine the possibility of synthesis-coupled substrate discrimination. Utilizing a rapid quench flow instrument, we monitored the +1 nucleotide elongation of 19 nt RNA primer by PriSLX in reactions activated by GTP or dGTP delivery, corresponding to RNA and DNA synthetic mode, respectively. As steady-state analyses, the reactions first proceed by activation of preformed polymerase-substrate complexes in the presence of excess substrate[21]. Importantly, as the template contains a mismatch to (d)GTP incorporation at the +2 position, under the short time frame,

PriSLX must disengage from +1 (d)GMP incorporation product and re-engage on new, unutilized substrate to increase product yield beyond the initial round of single-nucleotide extension.

For both the PriSLX RNA and DNA synthetic mode analyses, our data fit well to the burst-steady state model, implying a short, fast burst phase followed by a longer, slower steady-state phase as could be expected for a nucleic acid polymerase[21,23–25] (Fig. 5b). The burst phase is described by initial rate $(K_{obs})$[21,23–25]. Whereas, the steady-state phase determines the steady-state rate $(K_{ss})$, or the rate of turnover to new substrate after product formation[21,23–25].

Further highlighting how active PriSLX is as a DNA polymerase, the DNA synthetic mode initial rate $(K_{obs} = 2.86 \pm 0.58\,\text{s}^{-1})$ was ~8-fold faster than RNA synthetic mode initial rate $(K_{obs} = 0.36 \pm 0.10\,\text{s}^{-1})$ as shown in Fig. 5b. Given that the initial rate represents the chemistry of phosphodiester bond formation[21,23–25], we can conclude that PriSLX incorporates dNMP faster than NMP in the elongation of 19nt RNA primer.

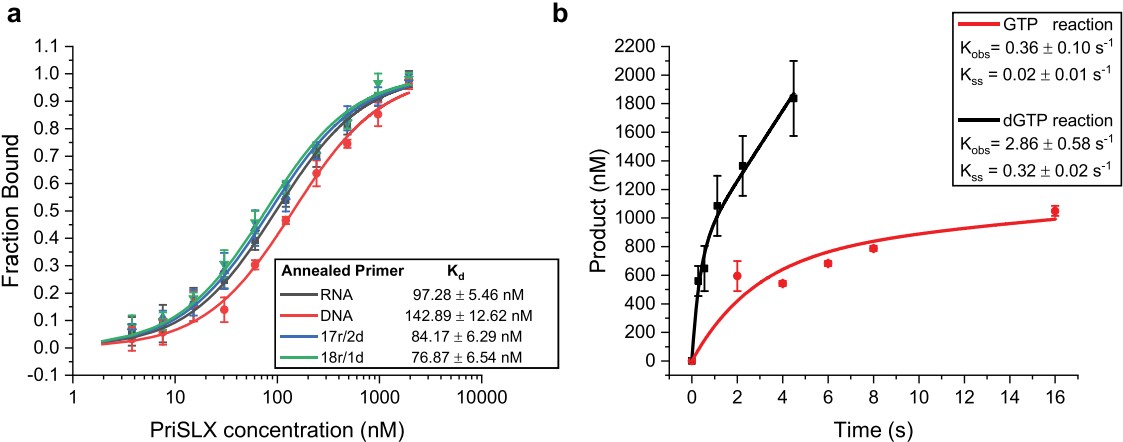

**Fig. 5 Binding and kinetic analyses of PriSLX. a** $K_d$ determination by fluorescence polarization. Titration series of PriSLX as indicated with indicated 5'-cy5-labeled primer annealed to 80 nt template. Data points indicate the mean and error bars represent standard deviation. Four independent replicates of this experiment were performed. **b** Steady-state kinetic analysis of PriSLX to determine initial rate of product formation ($K_{obs}$) and steady-state turnover rate ($K_{ss}$). Single nucleotide elongation reactions with GTP or dGTP as indicated and 5'-cy5-labeled 19 nt RNA-primer annealed to 80 nt template. Reactions timed by rapid quench flow as indicated; 60 °C. Data points indicate the mean and error bars represent standard deviation. Two independent replicates of this experiment were performed.

Even more interestingly, the DNA synthetic mode steady-state rate ($K_{ss} = 0.32 \pm 0.02$ s$^{-1}$) was ~16-fold faster than the RNA synthetic mode steady-state rate ($K_{ss} = 0.02 \pm 0.01$ s$^{-1}$) as shown in Fig. 5b. Importantly, studies of other nucleic acid polymerases show that the steady-state rate, a description of turnover, is dominated by the disengagement of polymerase from incorporation product[23,24]. Therefore, this ~16-fold difference in steady-state rate leads us to conclude that PriSLX more quickly disengages from the primer-template junction after the incorporation of dNMP than it does when a NMP is incorporated. Thus, despite similar binding affinities to primer-template substrates irrespective of the nature of the 3′ end (Fig. 5a), PriSLX exerts a striking synthesis-coupled substrate discrimination (Fig. 5b). Acting as a DNA polymerase, PriSLX is quicker to disengage and release the primer than when acting as an RNA polymerase.

This stark difference in disengagement provides a simple mechanistic explanation for how PriSLX terminates synthesis and releases 3′-DNA tailed hybrid primer for hand-off to and elongation by PolB1-HE. After dNMP incorporation commits PriSLX to DNA synthetic mode, PriSLX disengages from the primer-junction soon after—as PriSLX is quicker to disengage when acting as a DNA polymerase. Once PriSLX has disengaged, the DNA-terminating primer is a suboptimal substrate for PriSLX to re-engage with to resume elongation. Instead, this hybrid primer is optimal for recognition and elongation by PolB1-HE—thus PriSLX disengagement as a DNA polymerase facilitates the primer hand-off. By way of contrast, when PriSLX acts solely as an RNA polymerase, it would not only synthesize a non-optimal substate for PolB1-HE, but also be more prone to remain engaged on the primer, thus sterically preventing PolB1-HE from accessing the 3′-terminus.

**Hybrid primer synthesis by PriSLX may be triggered stochastically.** We have demonstrated that PriSLX requires both RNA synthetic capability (Fig. 1b) and DNA synthetic capability (Figs. 2 and 3) to optimally stimulate elongation by PolB1-HE. Furthermore, we have demonstrated that, at the single nucleotide incorporation level, PriSLX possesses DNA polymerase activity greater than its RNA polymerase activity (Figs. 4b, c, 5b). While in our biochemical assays we have been able to parse out the

effects of ribo- and deoxyribonucleotides by withholding one or another, in the cell, both substrates are present.

Previously, we have determined relative nucleotide concentrations in *Saccharolobus* cells[26], and so we calculate that per triphosphate-base on average, 2.4% of each base is in deoxyribo-form and 97.6% of each base is in ribo-form (Supplementary Fig. 3a). Further, we have measured the competition of ATP versus dATP for binding to the primase catalytic site[12]. Given that we have no reason to believe that ATP and dATP binding are not representative of all the base-nucleotides, we determine that the catalytic site has a 6.5-fold higher affinity for dNTP over NTP (Supplementary Fig. 3b). Additionally, with our new data, we consider +1 nucleotide elongation by PriSLX from a RNA-chain as a discrete, unbiased, 2-outcome incorporation event (NMP or dNMP is incorporated) because only when the preceding nucleotide is NMP does PriSLX readily extend the primer by NMP or dNMP incorporation (Fig. 4c—lanes 1–4 and 17–20).

Equally weighting the parameters of relative abundancy and binding affinity while considering +1 nucleotide elongation from a RNA-chain as a discrete, unbiased, 2-outcome incorporation event, we estimate that for each incorporation event lengthening a RNA-primer there is a 14% chance that the incorporated nucleotide is dNMP.

If we assume, as is congruent with our data (Fig. 1b), that the initiating nucleotide is a NTP and treat the incorporation events that follow as discrete events, we can readily build a cumulative geometric probability distribution[27] of first dNMP incorporation as a function of primer length (Fig. 6a). Modeling successive incorporation events until the first dNMP incorporation, our cumulative probability distribution estimates a ~94% probability of first dNMP incorporation by a primer length of 20 nucleotides or shorter (Fig. 6a). Notably, a primer length of 20 nucleotides corresponds to roughly the maximum length PriSLX can accommodate in vitro as we have previously demonstrated[12] and have confirmed in the current work (Fig. 4b).

Furthermore, considering that PriSLX commits to DNA synthesis after dNMP incorporation (Fig. 4c) and readily disengages from the primer when acting as a DNA polymerase (Fig. 5b), our cumulative distribution of first dNMP incorporation (Fig. 6a) implies a primer-length product range in agreement with the observed range of primer lengths detected in Okazaki fragments purified from archaeal cells[28]. Therefore, we propose

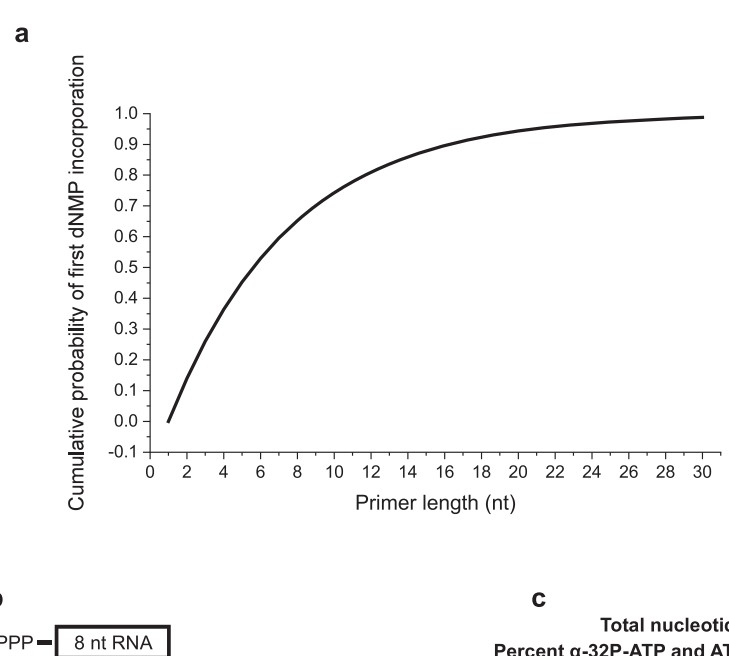

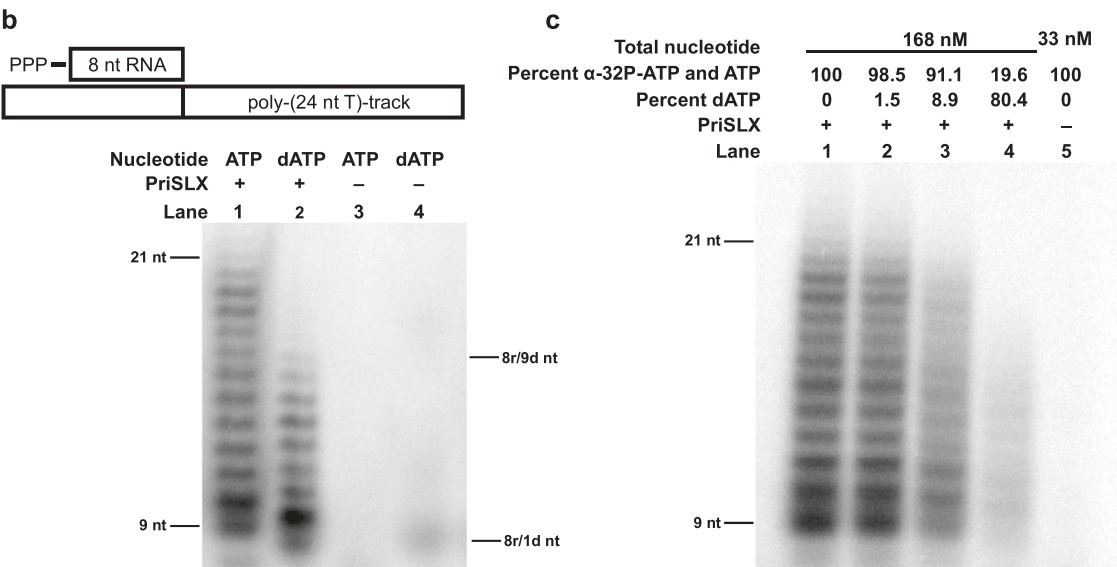

**Fig. 6 Impact of dNMP incorporation on primer synthesis. a** Geometric probability model of dNMP incorporation by PriSLX as a function of primer length. Model assumes NTP as the initiating nucleotide and that nucleotide incorporation proceeding from RNA-primer is governed exclusively by equal weighting of relative dNTP/NTP abundance and elongation-site binding affinity. **b** Top panel: representation of the 5′-triphosphate 8 nt RNA-primer annealed to the poly-(24 nt T)-track. Bottom panel: primer elongation reactions of ATP (2 µCi/20 ul α-$^{32}$P-ATP and 165 nM ATP) or dATP (2 µCi/20 ul α-$^{32}$P-dATP and 165 nM dATP) as indicated; 15 nM 5′-triphosphate 8 nt RNA-primer annealed to poly-(24 nt T)-track template; and 3.75 nM PriSLX as indicated. Reactions were incubated for 2 min at 75 °C prior to termination and electrophoresis on denaturing polyacrylamide gels. Four independent replicates of this experiment were performed. **c** Primer elongation reactions of 2 µCi/20 ul α-$^{32}$P-ATP with unlabeled ATP and dATP as indicated for total nucleotide concentration as indicated; 15 nM 5′-triphosphate 8 nt RNA-primer annealed to poly-(24 nt T)-track template; and 3.75 nM PriSLX as indicated. Reactions were incubated for 2 min at 75 °C prior to termination and electrophoresis on denaturing polyacrylamide gels. Three independent replicates of this experiment were performed.

that a stochastically-driven model can account for dNMP incorporation and the consequent synthetic mode switch.

**Nucleotide preference programs primer length**. Finally, we further tested PriSLX synthetic capabilities using a 5′-triphosphate 8 nt RNA primer annealed to a template adjacent to a poly-(24 nt T)-track (Fig. 6b). Using this substrate, we monitored primase-mediated primer extension by following incorporation of α-$^{32}$-P labeled ATP or dATP. The 5′-triphosphate allows for the full engagement of the caliper primer-length counting mechanisms that we have previously described[12] and the 8 nt RNA primer provides a dynamic range before elongation to the full-length permitted by the engaged caliper (~20 nt).

Initially, we revisited PriSLX-mediated polymerization in separate reactions for RNA synthetic mode and DNA synthetic mode, by incubation with ATP or dATP, respectively (Fig. 6a). As expected, the ATP-containing reaction produced overall product lengths of 21 nt maximum, in good agreement with our previous work[10,12]. The dATP-activated reaction produced shorter product lengths of ~17nt maximum, also in agreement with our previous work[12].

However, we now have greater insight into why there is this difference in product length between PriSLX acting as an RNA versus DNA polymerase. Acting as an RNA polymerase, PriSLX is slower to disengage from the elongation product (Fig. 5b) and if it were to disengage, re-engagement would likely be productive, as RNA-terminating substrate is optimal for PriSLX (Fig. 4c).

In contrast, acting as a DNA polymerase, PriSLX is faster to disengage from the elongation product (Fig. 5b) and re-engagement would likely be unproductive, as substrate with 3′-DNA is suboptimal for PriSLX (Fig. 4c). Thus, this results in the shorter product lengths we observe (Fig. 6b).

Finally, we tested the effect of increasing the relative concentration of dATP to ATP in elongation reactions (Fig. 5c). These reactions all contained a constant absolute amount of α-[32]P-ATP radiolabel and a constant total molar concentration of nucleotide. However, the molar concentrations of ATP and dATP, individually, were modulated so that, as a series of reactions, progressively more of the total nucleotide concentration would be dATP. As the radiolabel in each reaction is a constant absolute amount of α-[32]P-ATP, we would anticipate less incorporated radiolabel signal if dATP were to outcompete ATP. Further, given our insight into how PriSLX acts as a DNA-polymerase (Fig. 6b), outcompeting dATP should also concomitantly drive shorter product lengths due to PriSLX disengagement as the enzyme acts as a DNA polymerase (Fig. 5b), producing a product that is suboptimal for further extension by PriSLX (Fig. 4c) even if it were to re-engage.

Significantly, when we performed the dATP competition assay, our results were in line with what we would expect if dATP outcompeted ATP (Fig. 6c). Notably, the coupled decrease in incorporated radiolabel and decreased product length occurred when overall reaction dATP was between 1.5 and 8.9% of the total nucleotide in the reaction (Fig. 6c—lanes 2–3), a range that brackets the 2.4%/98.6% dNTP/NTP relative nucleotide abundance estimated in the *Saccharolobus* cell[26] (Supplementary Fig. 3a). Thus, this assay empirically demonstrates dNTP outcompeting NTP—despite being less abundant—for incorporation by PriSLX during nucleic acid elongation. Our data support the conceptual accuracy of the geometric probability model, which predicts that, during PriSLX synthesis of primer of physiologically relevant length under physiologically relevant nucleotide concentrations, a dNMP incorporation event is favored. (Fig. 6a).

## Discussion
Our data provide a mechanistic explanation for the long-standing observation that archaeal primases possess both DNA and RNA synthetic properties[13,14]. While this phenomenon has been well-established for many years, a number of features have been puzzling. In particular, it has been documented that the catalytic site of primase has a higher affinity for dNTPs yet generates shorter DNA products than RNA products[12]. Furthermore, the presence of dNTPs has been observed to have an inhibitory effect on total nucleic acid synthesis by primase[11,12]. Previously, we did not find evidence for the synthesis of long contiguous tracts of DNA preceded by contiguous RNA. However, those previous assays would not have resolved short tracts of 1–3 dNMPs appended to a RNA primer. Our current data, coupled with our previous measurements of relative nucleotide concentrations in vivo[26], allow us to integrate these aforementioned observations.

Our data support a model of archaeal primer hand-off from primase to replicative polymerase (Fig. 7). As primase elongates the primer, stochastic incorporation of a dNMP commits primase to DNA synthetic mode, resulting in synthesis of a short 3′-DNA tail. Acting as a DNA polymerase, primase readily disengages from the hybrid primer-junction, releasing an optimal substrate for elongation by the replicative polymerase thereby facilitating the primer hand-off.

Our model is also compatible with the recent report of the primase of *Pyrococcus abyssi* playing a role in lesion bypass[15]. When elongating from a 3′-DNA-terminus, the primase would be committed to DNA synthetic mode. Thus, acting as a DNA polymerase, the primase would be prone to disengage quickly. As primase lacks proof-reading activity, this would limit the length of the primase-synthesized DNA tract and thus minimize the likelihood of introducing mismatches during the repair process.

We note that, in agreement with our findings, the *Pyrococcus* enzyme also appears to be more efficient as a DNA polymerase[29,30]. Intriguingly, however, DNA-primer extension assays performed over extended time frames resulted in significantly longer DNA products in the *Pyrococcus* system than we observe with *Saccharolobus*. Furthermore, extension of a DNA primer by NTPs appears highly length-constrained in the *Pyrococcus* system. Given that *Pyrococcus*, unlike the crenarchaeal *Saccharolobus*, utilizes a D-family polymerase for genome replication, it will be of considerable interest to perform a detailed examination of substrate preference and requirements for primer hand-off in a euryarchaeal system.

We have previously proposed a caliper model for primer length determination, supported by the ongoing interaction of the 5′-triphosphate of the primer with the PriX subunit during primer elongation[12]. Our current work is fully compatible with that model, and we speculate that the caliper may function as a back-up pathway to constrain primer length should a dNMP incorporation event fail to take place (Fig. 7c). Indeed, we emphasize that the experimental evidence for the caliper model was derived from data generated with the primase in RNA synthetic mode[12].

Finally, as noted above, primases lack proofreading activity and utilizing primase to generate the first DNA nucleotides on each Okazaki fragment might appear to introduce a potential source of replication-associated errors. However, two mechanisms likely prevent such mutagenesis. First, a mismatch at the 3′ end of a primer may prevent productive engagement of DNA polymerase or could act as a substrate for error-correction by the DNA polymerase's proof-reading activity. Second, by limiting DNA synthesis to only a short stretch of nucleotides, it seems plausible that any mismatch generated by misincorporation event(s) would sufficiently destabilize the 5′-end region of DNA of an Okazaki fragment such that the destabilized region would serve as a substrate for Fen1 during Okazaki fragment maturation, particularly so in an organism such as *Saccharolobus* that grows between 75 and 80 °C. Thus, as PolB1-HE, in the context of the PCNA-Lig1-Fen1-PolB1 "Okazakisome" complex, completes synthesis of the upstream Okazaki fragment, it would correct any primase-generated errors during the Okazaki fragment maturation process[21,31,32].

## Methods
**Protein expression and purification.** Expression and purification of recombinant wild-type proteins were essentially as described for PolB1-HE (his-tagged)[21] and PriSLX wild-type[10], with the exceptions that all IPTG-inductions were performed for 4 h after 2-h pre-induction growth, all lysis were done by French Press or sonication, and the final PriSLX preparations were supplemented with 1 mM DTT prior to storage at −80 °C.

Recombinant PolB1-HE polymerase-dead (his-tagged) and PriSLX polymerase-dead mutants were expressed and purified as their wild-type counterparts, with the substitution of mutant expression vectors. PolB1 polymerase-dead mutant (D655A; D657A) expression vector was generated by single-primer (5′-[phos] AGAAGGT TTAACTGTATTATACGGTGC TACTGCTTCTTTATTCCTCCTTAATCCTCC C-3′) site-directed mutagenesis of PolB1-encoding *pET33b* vector[21], and the sequence construct was confirmed by Sanger sequencing (Genewiz). The PriSL polymerase-dead, catalytic mutant vector encoding mutations *PriS* D101A and D103A was previously described[33]

**In vitro replication assay.** In vitro replication was reconstituted similarly to previously described[10]. 20 µl reactions of 200 µM NTPs (when indicated); 30 µM dNTPs; 2 µCi/10 µL α-[32]P-dATP; 25 nM M13 ssDNA template; and 25 nM of protein(s) as indicated were assembled in 1× reaction buffer (50 mM Tris pH 6.0; 10 mM MgCl₂; 10 mM DTT; and 0.1 mg/ml BSA). Reactions were incubated for 30 min at 75 °C. Reactions were quenched, resolved by 1% alkaline gel electrophoresis, and visualized

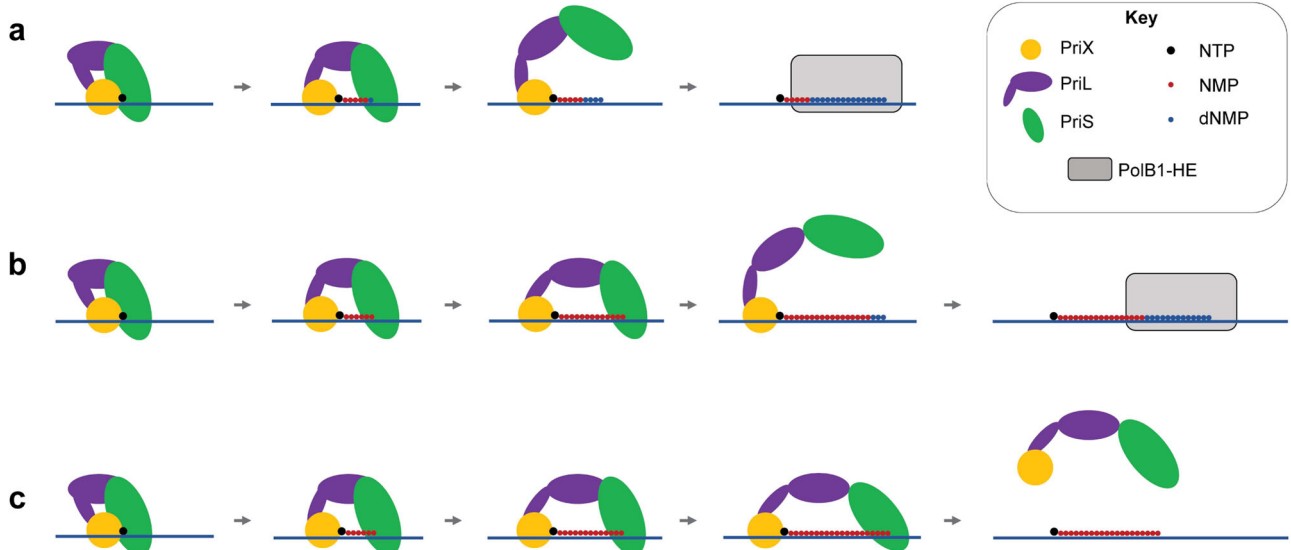

**Fig. 7 Model of the primer hand-off.** After primase initiates primer synthesis by RNA-synthesis, a favored stochastic deoxyribonucleotide incorporation switches primase to DNA-synthesis. This promotes primase disengagement from the primer and subsequent hand-off to replicative polymerase. **a**, **b** A scenario of earlier or later switch to DNA-synthesis, respectively, by primase. **c** While unfavored, a scenario where primase does not switch to DNA-synthesis. As a back-up failsafe, the caliper mechanism constrains RNA-primer length, halting synthesis. With synthesis halted, primase dissociates from the primer.

by phosphorescence imaging similarly to previously described[10]. Plus/minus polymerase-active assay was done in duplicate and plus/minus NTPs assay was done in triplicate. Ladder overhangs were labeled by Klenow radiolabel end-filling.

For product quantification as indicated in text, gel electrophoresis trace files were generated in ImageJ (version 1.53a; NIH) and analyzed in OriginPro 2018 (version SR1-b9.5.1.195; OriginLab Corporation). Mean fold-change and population standard deviation was determined, treating each replicate independently.

**Primer-template annealing**. For elongation and kinetic assays, the appropriate primer (Table S1) was mixed with 1.5× molar excess of template in 1× annealing buffer (10 mM Tris-HCl pH 6.8, 100 mM KCl, and 10 mM MgCl₂). The solution was brought to 100 °C and then slowly cooled by PCR thermal-cycler. When applicable, fluorescent, Cy5-labeled-substrate concentrations were confirmed by absorbance at 650 nm. For binding assays, the appropriate primer (Table S1) was mixed with template 1:1 and otherwise treated the same.

**19 nt primer elongation assays**. 20 µl reactions were assembled and incubated similarly to previously described[21]. Reactions of 100 µM NTPs (as indicated); 100 µM dNTPs (as indicated); 25 nM 5′-cy5-labeled primer as indicated annealed to 80 nt template; PolB1-HE (as indicated); PriSLX (as indicated); polymerase-dead variants (as indicated); and fresh BSA (as indicated) were assembled in 1× reaction buffer (50 mM Tris-HCl pH 6.8; 40 mM KCl; 10 mM MgCl₂; 10 mM DTT; and 0.1 mg/ml BSA). Reactions were incubated for 5 min, unless otherwise for an indicated time course; reactions were incubated at 75 °C. Reactions were then quenched and analyzed by polyacrylamide (12% gel) electrophoresis as previously described[21]. All assays were done in at least triplicate.

**Fluorescence polarization**
*PolB1-HE binding.* Fluorescence polarization was done similarly to previously described[12,21]. 100 µl reactions of 10 nM Cy5-labeled primer-template as indicated and PolB1-HE concentrations as indicated were assembled in binding reaction buffer (11 mM Tris pH 6.0; 100 mM NaCl). Before reaction assembly, PolB1-HE was treated with 1.56 mM EDTA per 1 nM PolB1-HE. In the Synergy Neo 2 plate reader integrated with Gen5 (version 3.0019; BioTek) workflow, reactions were incubated at 60 °C for 10 min prior to fluorescence polarization detection at the same temperature (excitation at 620 nM/40 nM; emission at 680 nM/30 nM). For each primer-template as indicated, the mean of 4 replicates were fit to the single-site binding model[21] on OriginPro 2018 (version SR1-b9.5.1.195; OriginLab Corporation).

*PriSLX binding.* Fluorescence polarization was done similarly to previously described[12,21]. 100 µl reactions of 10 nM Cy5-labled primer-template as indicated and PriSLX concentrations as indicated were assembled in reaction buffer (50 mM Tris-HCl pH 6.8; 40 mM KCl; 10 mM MgCl₂; 10 mM DTT; and 0.1 mg/ml BSA). In the Synergy Neo 2 plate reader integrated with Gen5 (version 3.0019; BioTek)

workflow, reactions were incubated at 60 °C for 5 min prior to fluorescence polarization detection at the same temperature (excitation at 620 nM/40 nM; emission at 680 nM/30 nM). For each primer-template as indicated, the mean of 4 replicates for wild-type or 3 replicates for polymerase dead were fit to the single-site binding model[21] on OriginPro 2018 (version SR1-b9.5.1.195; OriginLab Corporation).

**Steady-state analysis by rapid quench flow**. Burst-steady-state kinetic analysis by rapid quench flow instrument (KinTek RQF-3) was performed similarly to previously described[21]. Reactions were initiated by mixing reactant solution containing PriSLX and Cy5-labeled 19nt RNA primer-template with reactant solution containing (d)GTP as indicated for in-reaction concentrations of 770 nM PriSLX, 3.5 µM primer-template, and 100 µM (d)GTP as indicated in 1x reaction buffer (50 mM Tris-HCl pH 6.8; 40 mM KCl; 10 mM MgCl2; 10 mM DTT; and 0.1 mg/ml BSA). Reactions were incubated at 60 °C. At the reaction times indicated (2–16 s for GTP reaction series; 0.28 to 4.48 s for dGTP reaction series), the reactions were quenched by mixing with 50 mM EDTA for a final EDTA concentration of ~20 mM. Quenched reactions were recovered with an additional 300 µl expulsion volume.

For each reaction, the recovered volume was mixed with 1x loading dye (8 M urea, 0.01 M EDTA, 1× TBE, and 1× bromophenol blue) in a ratio of 5:100 prior to separation and visualization by polyacrylamide (12% gel) electrophoresis essentially as previously described[21]. Gel electrophoresis trace files were generated in ImageJ (version 1.53a; NIH). Substrate and product signals were then analyzed in OriginPro 2018 (version SR1-b9.5.1.195; OriginLab Corporation). For each reaction series (dGTP and GTP reactions), the product mean and population standard deviation for each reaction time was calculated from two replicates.

Each averaged series (dGTP and GTP) was plotted as product (y-axis) versus time (x-axis). In OriginPro 2018 (version SR1-b9.5.1.195; OriginLab Corporation), each series was regressed to the burst-steady-state model similarly to previously described[21].

$$[\textbf{product}] = A(1 - \exp(-k_{obs}t)) + k_2t \qquad (1)$$

Given an enzyme concentration ($A$) of 770 nM, the initial rate of product formation was defined as ($k_{obs}$). Additionally, steady-state turnover rate ($k_{ss}$) was derived.

$$\textbf{K}_{ss} = \textbf{k}2/\textbf{A} \qquad (2)$$

**Cumulative geometric probability distribution**
*Relative nucleotide abundance.* Taking our previously published nucleotide concentrations in *Saccharolobus* cells[26] (Supplementary Fig. 2a), we calculated the unweighted mean of relative deoxyribo-form and ribo-form abundance per triphosphate-base.

*Relative NTP and dNTP binding affinity to PriSLX elongation site.* We took our previously published anisotropy binding competition of ATP versus dATP for binding to the primase catalytic site, displacing fluorescently-labeled ATP[12]

(Supplementary Fig. 2b). We plotted polarization response (y-axis) to (d)ATP titrant concentration (x-axis) and regressed the plotted data to the logistic function in OriginPro 2018 (version SR1-b9.5.1.195; OriginLab Corporation).

$$y = A_2 + (A_1 - A_2)/(1 + (x/x_0) \wedge p) \tag{3}$$

For each titrant, ($x_0$) defines ($EC_{20}$), and we took ATP and dATP relative binding as representative of all NTPs and dNTPS, respectively.

*Probability of dNMP incorporation per discrete incorporation event and the geometric cumulative probability distribution.* Assuming unbiased elongation by PriSLX with either nucleotide from an RNA-primer, the odds for and against dNMP incorporation proceeding form an RNA-primer were calculated per a single, independent incorporation event, defining the odds for NMP incorporation as the odds against dNMP incorporation. Relative abundance and relative binding affinity were weighted equally.

$$\frac{\text{Odds for dNMP incorporation}}{\text{Odds against dNMP incorporation}} = \frac{\text{dNTP abudance}}{\text{NTP abudance}} \times \frac{1/\text{dNTP EC50}}{1/\text{NTP EC50}} \tag{4}$$

Odds for and against dNMP incorporation per a single, independent incorporation event proceeding from an RNA-primer were then converted to probability of dNMP incorporation per a single, independent incorporation event proceeding from an RNA-primer ($p$).

$$p(\text{dNMP incorporation}) = \frac{\text{Odds for dNMP incorporation}}{\text{Odds for + against dNMP incorporation}} \tag{5}$$

Assuming PriSLX initiates primer synthesis with triphosphate nucleotide and lengthens the primer nucleotide length ($l$) through discrete, independent, single nucleotide incorporation events ($x$), then the integer of incorporation events ($x$) was described in terms of the integer of nucleotide primer length ($l$).

$$x = l - 1 \tag{6}$$

Therefore, assuming primer synthesis is initiated by a single NTP, geometric probability[27] of first dNMP incorporation ($P$) as a function of incorporation events ($x$) was determined and converted into a function of nucleotide primer length ($l$).

$$P(x) = ((1 - p_{dNMP\ incorporation}) \wedge (x - 1)) \times p_{dNMP\ incorporation} \tag{7}$$

$$P(l) = ((1 - p_{dNMP\ incorporation}) \wedge (l - 2)) \times p_{dNMP\ incorporation} \tag{8}$$

And thus, cumulative geometric probability[27] of first dNMP incorporation ($F$) as a function of primer length ($l$) was modeled.

$$F(l) = \sum ((1 - p_{dNMP\ incorporation})^{l-2}) \times p_{dNMP\ incorporation} \tag{9}$$

### 5′-triphosphate 8 nt RNA primer radiolabeled elongation assay

*ATP or dATP reaction assay.* Similar to previously described[12], 20 μl primer elongation reactions of ATP (2 μCi/20 μl α-$^{32}$P-ATP and 165 nM ATP) or dATP (2 μCi/20 μl α-$^{32}$P-dATP and 165 nM dATP) as indicated; 15 nM 5′-triphosphate 8 nt RNA primer annealed to poly-(24 nt T)-track template; and 3.75 nM PriSLX as indicated were assembled in 1x reaction buffer (50 mM Mes-NaOH pH 6.0; 10 μM ZnCl$_2$; and 0.1 mg/ml BSA).

*dATP titration assay.* Similar to previously described[12], 20 μl primer elongation reactions of 2 μCi/20 μl α-$^{32}$P-ATP; ATP and dATP as indicated for total nucleotide concentration as indicated (168 or 33 nM); 15 nM 5′-triphosphate 8 nt RNA primer annealed to poly-(24 nt T)-track template; and 3.75 nM PriSLX as indicated were assembled in 1× reaction buffer (50 mM Mes-NaOH pH 6.0; 10 μM ZnCl$_2$; and 0.1 mg/ml BSA).

Both assay reactions were incubated for 2 min at 75 °C. Reactions were then quenched, resolved by polyacrylamide (18% gel) electrophoresis, and visualized by phosphorescence imaging essentially as previously described[12]. All assays were performed in at least triplicate.

**Reporting summary**. Further information on research design is available in the Nature Research Reporting Summary linked to this article.

## Data availability

The data and reagents that support the findings of this study are available from the corresponding author upon reasonable request. Source data are provided with this paper.

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

## Acknowledgements
Work on DNA replication in SDB's lab is supported by The College of Arts and Sciences, Indiana University and NIH Grant 1R01GM125579-01. We thank members of the Bell Lab, Rachel Samson, Jade Katinas and Charles Dann III at Indiana University, and KinTek.

## Author contributions
M.D.G. designed experiments, performed experiments, analyzed data and wrote the paper. J.D.D. performed the mathematical modeling and contributed to the text of the manuscript. S.D.B. designed experiments, analyzed data and wrote the paper.

## Competing interests
The authors declare no competing interests.
