## [Peer Review File · Nature Communications]

REVIEWER COMMENTS

Reviewer #1 (Remarks to the Author):

This work is an extension of their previous novel findings describing the *Sulfolobus solfataricus* primase. Previously, this group discovered that the *Sulfolobus* primase complex, PriSLX, acts as a caliber where PriX binds the 5' end while PriS synthesizes the primer. A long standing question is what limits the primase length and how primase hands over the primer to the polymerase. The authors demonstrate in this work that the *Sulfolobus* polymerase, PolB1-HE is inefficient in extending an RNA primer, but inclusion of PriSLX stimulated DNA extension by PolB1-HE. The authors demonstrate that is due to the incorporation of dNTPs into the primer by the PriSLX. In this study, the authors have demonstrated that PriSLX can incorporate dNTPs and when it does, it readily disengages providing an opportunity for the polymerase, PolB1-HE to take over to synthesize DNA.

Comments:

I would really like to see the normal length distribution of primase products from a de novo primase reaction, not from an extension of a preformed RNA primer/DNA template. Can the authors perform a de novo primase reaction with ³²P-alpha-dNTP to see the typical length of product with a DNA tail?

Does the PriSLX need to synthesize a certain length RNA before switching to DNA synthesis? Or is the dNMP incorporation by PriSLX entirely stochastic? What is the minimal length RNA primer needed before a dNMP is observed to be incorporated?

The authors mention that the limit of addition of dNMPs to the primer to 3 nucleotides or less limits the possibility of mismatches. This is important because it is well known that primases have low fidelity. The incorporation of DNA into the primer begs the question of what is the efficiency of the Pol1B-HE in extending a primer with a DNA mismatch at the 3' end?

Can the authors speculate/discuss whether this model holds true for the mammalian primase and the handoff to Pol alpha. A comparison of similarities would be welcome.

Some of the data/figures can be moved to the supplemental material. Fig 7 is a probability curve that should not be in the main text.

Reviewer #2 (Remarks to the Author):

The authors report that the DNA polymerase of *Sulfolobus solfataricus* PolB1-HE extends an RNA primer with significantly lower efficiency in comparison to a DNA primer, and the DNA-polymerizing activity of primase PriSLX significantly stimulates RNA primer extension by PolB1-HE. Based on a stochastic mode of the first dNMP insertion into an RNA primer, PriSLX affinity for ATP and dATP, and *in vivo* ratios of rNTPs and dNTPs, it was estimated that one dNMP should be incorporated into the 20-mer primer on average. The authors presented some evidence that after attachment of one dNMP to the 3'-end of an RNA primer, PriSLX switches into DNA polymerizing mode and generates a DNA tail, which promotes primer release by primase and its hand off to PolB1-HE. In Discussion section, it was speculated that the DNA tailing is the main mechanism of primer synthesis termination by PriSLX, while the previously proposed mechanism of RNA synthesis termination based on primer length (Yan et al., 2018 PNAS) serves as a backup.

The main and critical drawback of this work is comparison of RNA and DNA synthetic activities of PriSLX at physiologically non-relevant relative concentrations of rNTPs and dNTPs. For example, experiments presented in Figure 5 and intended to show the higher efficiency of PriSLX in DNA polymerizing mode versus the RNA polymerizing one were conducted at the same concentration of rNTPs and dNTPs (0.1mM). On the other hand, the authors mention that concentration of dNTPs comprises only 2.4% of the total nucleotide triphosphate pool in *Sulfolobus solfataricus*. Moreover, in the previous report of this group (Yan et al., 2018 PNAS), performed at physiologically relevant relative concentrations of nucleotide triphosphates, it was observed that PriSLX functions as an RNA synthetic enzyme. In particular, they wrote **“did not find any compelling evidence for an ordered sequential synthesis of first RNA and then DNA by the primase”**.

Actually, no evidence is provided that PriSLX switches into DNA polymerizing mode after attachment of one dNMP to the 3'-end of an RNA primer. In fact, Figure 5C shows that DNA polymerase activity of primase dramatically reduced on 18r/1d primer (lanes 29-31) in comparison to RNA primer (lanes 17-19). So, if to correctly interpret Figure 5C, it shows that addition of one dNMP to the RNA primer severely inhibits the following primer extension by PriSLX with ribo and deoxy NTPs. This is a very inefficient way to generate a DNA tail of 2-3 dNTPs.

The other claim that dNMP incorporation by PriSLX drives primer disengagement is not supported. There is no clear description what the term “disengagement” means and what value characterizes it, the dissociation constant (Kd) or the dissociation rate constant (Koff). The binding studies show similar Kd values for DNA, RNA and chimeric primers. In this case, Koff values for chimeric primers could be higher in comparison to RNA primer if dissociation rate constant (Kon) values are also increased but there is no clear reason for this. Kon is defined by diffusion rates for primase and substrate and depends on their size. If authors believe that DNA tailing increases Koff value, they should directly measure it for RNA, DNA and chimeric primers. It was mentioned that primer disengagement somehow connected to the steady-state rate but no clear explanation and/or equation were provided. In addition, if incorporation of just one dNMP into the RNA primer promotes primer disengagement, there is low probability that PriSLX would be able to incorporate a second and third dNMP in the presence of PolB1-HE. Thus, proposed disengagement mechanism contradicts with DNA tailing mechanism.

There are some issues with kinetic studies. Reactions with time points until 16 sec should be analyzed in the case of dGTP. It is important for the accuracy of burst assay that at reaction start an enzyme is saturated with substrates, which should be supplied at concentration ten times higher than corresponding Kd value. The RNA primer was supplied at concentration

only 2.5 times higher versus the K_d value. A K_d value for PriSLX/GTP complex was not provided so it is not clear whether PriSLX was saturated with it. It is important that PriSLX affinity for substrates is measured at the same conditions (buffer, salt, temperature), which are used in kinetic assay. If doing these measurements is not feasible, it would be satisfactory if the kinetic studies will be repeated at next concentrations: 10μM primer, 2.5μM PriSLX, 0.5mM (d)GTP.

Interpretation of results presented on Figure 8 is misleading. Elongation of a 5'-triphosphate RNA-primer by PriSLX shows clear synthesis termination at 10-mer and not at 20-mer as authors concluded. The intensity of a 10-mer band is several fold higher versus the other bands (lanes 1 and 2). Moreover, the intensity of bands >10-mer is overestimated because the α-³²P-(d)ATP and the template with poly(dT) track were used in reaction resulting in higher specific radioactive activity for next extension product versus the previous one. For example, the intensity of a 20-mer product is overestimated 6 times versus the 10-mer. For synthesis termination studies, it is important to label the primer only at one site in order to observe the true distribution of extension products. It is not clear why the authors decided do not use the same template as in previous study (Yan et al., 2018 PNAS), which has a poly(dA) track with one thymine and allows incorporation of only one α-³²P-(d)AMP into the primer. In addition, the level of products with length of >10 nucleotides would be further reduced in the presence of PolB1-HE, which will bind the 10-mer primer and prevent its extension by primase.

Moreover, slow extension of RNA primer by PolB1-HE with the first dNMP might have a biological sense. Is it a unique feature of PolB1-HE or there are known examples of other DNA polymerases, which take the primer from primase and demonstrate the same discrimination against the RNA primer? Affinity of PolB1-HE for RNA, DNA and chimeric primers should be analyzed. If discrimination against the RNA primer is based on affinity, it might be not the issue *in vivo* where the local substrate concentration shall be high enough to saturate PolB1-HE.

Reviewer #3 (Remarks to the Author):

The manuscript written by Greci MD et al is about biochemical analysis of the DNA replication mechanism in the hyperthermophilic archaeon, and described that PolB1-HE cannot utilize the 3'-RNA end to extend, but when PriSLX incorporates dNTP into the RNA end of the primer, PolB1-HE becomes to extend it. PriSLX easily dissociates from the 3'-DNA end, and therefore, a primer can be hand-off from PriSLX to PolB1-HE for further processive DNA synthesis during the DNA replication process. It is the first report describing the transferring mechanism from primase to replicase during the replication process in Archaea, and I think it is generally interesting for the readers of Nature Communication. I find several parts that I'm not completely satisfied with the descriptions by the authors. I hope the following comments will help to improve the manuscript before publication.

line 25. "vertebrates possess an AEP family member, PrimPol, that acts - - -"
I think PrimPol is also found from Archaea, bacteria, and virus. Revise this description precisely.

line 64. "using purified recombinant Sulfolobus PriSLX and the replicative B-family DNA polymerase holoenzyme, PolB1-HE."
I imagine from the surrounding description and citations that Sulfolobus seems to be *S. solfataricus*, but no description is found anywhere in the manuscript which organism was used in this study. Clarify it. In addition, Sulfolobus solfataricus has been recharacterized as 'Saccharolobus'solfataricus. It may be better to be updated.

line 118. PriSLX PD is first mentioned here, but it is already included in Figure 1. Please check the descriptions and figure numbers again in the whole manuscript.

Figure 3a, lane 1-3, and 17-19 were the results from exactly same conditions as those of Figure 2c lane 1-3 and 4-6 (or Figure 2a, lane 17-19 and 20-22). However, they do not look the same. I think the consistency between the figures in the same manuscript is the essential issue for the good paper.

The description for Figure 3b is missing in the text. The result of RNA primer-extension (Fig 3b) is clearly different from hybrid primer-extension (Fig 3a). BSA and PriSLX PD non-specifically promote the extension by PolB1-HE from the hybrid primer, but they do not look like promoting the RNA primer-extension. Clarify this point.

Related to the above comment, "very modest enhancement" in line 128 is really weak.
I think the result in Figure 3 may be enough to be shown in a Suppl figure.

Figure4, lane17-20. Why does the extension of 19 nt DNA, but not others, can be promoted by BSA? Some reasonable explanation is required.

line 170-174 "Remarkably, even though primers in our current assays possessed a 5'-florescent label, and thus are presumably not recognized by PriX, an overwhelming majority of 19nt RNA primer elongation remained at +3 nucleotides (22 nt product) for both RNA and DNA synthetic modes (Figure 5b - lanes 10 and 20), suggesting that structural aspects of length constraint remain engaged despite the fluorescent label."

The authors' group published in their previous work that PriSLX containing PriX initiation site mutations, D70A, R72A or R74A, which lost the interaction with the 5'-end of the primer, lost the function of the caliper and the primer length was not limited in the Fig 3B of Yan et al., 2018, PNAS. However, in this study, they showed the data that PriSLX functions as a caliper without the interaction of PriX and 5'phosphate (Fig. 5B lane 8-10). Clarify this inconsistency and add some data to explain the result of Fig 5B.

line 353. "Our model is also compatible with the recent report of the primase of *Pyrococcus abyssi* playing a role in lesion bypass¹⁵. When polymerizing from a 3'-DNA-terminus, the primase would be committed to DNA synthetic mode. Thus, acting as a DNA polymerase, the primase would be prone to disengage quickly. As primase lacks proof-reading activity, this would limit the length of the primase-synthesized DNA tract and thus minimize the likelihood of introducing mismatches during the repair process."

I know that the research group published the paper of ref 15 also reported the results of a primer extension performed using *P. abyssi* PriSL and d/rNTP in a different paper. The synthesized product of dNTP alone is longer than that of rNTP alone or d/rNTP mixture in the report (*J. Mol Biol* 430, 4908-4924), which is the opposite of the results from *S. solfataricus*. Combined with other earlier studies of *P. furiosus* (*J Biol Chem* 276, 45484, 2001; *Nucleosides Nucleotides and Nucleic Acids*, 25, 681, 2006), I wonder if the presented model in this study could not be adapted to Euryarchaeota. More pondered discussion could be required if the conclusion is compatible to other archaea. To confirm if the hypothesis proposed by the authors (Figure 9a-b) is true in *Sulfolobus*, I think the experiment using PriSLX initiation-site mutants (PriX D70A, R72A or R74A) should be performed and see if the caliper function does not actually work. I would like to ask the authors to do this experiment.

Fig 5C right side. Typo "+3 dMPs"

line 202. Typo "yield pf product"

line 244. Typo "it dies when"

Figure 8b. Explain "ATP*" in the legend.

Table S1. The 3'-end of 17r/2d should be rACG, not rArCG.

Reviewer #1 (Remarks to the Author):

This work is an extension of their previous novel findings describing the *Sulfolobus solfataricus* primase. Previously, this group discovered that the *Sulfolobus* primase complex, PriSLX, acts as a caliber where PriX binds the 5' end while PriS synthesizes the primer. A long standing question is what limits the primase length and how primase hands over the primer to the polymerase. The authors demonstrate in this work that the *Sulfolobus* polymerase, PolB1-HE is inefficient in extending an RNA primer, but inclusion of PriSLX stimulated DNA extension by PolB1-HE. The authors demonstrate that is due to the incorporation of dNTPs into the primer by the PriSLX. In this study, the authors have demonstrated that PriSLX can incorporate dNTPs and when it does, it readily disengages providing an opportunity for the polymerase, PolB1-HE to take over to synthesize DNA.

Comments:

I would really like to see the normal length distribution of primase products from a de novo primase reaction, not from an extension of a preformed RNA primer/DNA template. Can the authors perform a de novo primase reaction with ³²P-alpha-dNTP to see the typical length of product with a DNA tail?

An analogous experiment is described in Yan et al, PNAS, Figure 2B, at a physiologically relevant ratios of dATP/ATP (lanes 8 and 9) we observed primers principally between 9 and 20nt in length.

Does the PriSLX need to synthesize a certain length RNA before switching to DNA synthesis? Or is the dNMP incorporation by PriSLX entirely stochastic? What is the minimal length RNA primer needed before a dNMP is observed to be incorporated?

At this point we do not know the answer to this question. However, the modeling that we present supports an entirely stochastic incorporation, governed by the relative balance of NTPs and dNTPs and the affinity of the primase for these two different classes of substrate.

The authors mention that the limit of addition of dNMPs to the primer to 3 nucleotides or less limits the possibility of mismatches. This is important because it is well known that primases have low fidelity. The incorporation of DNA into the primer begs the question of what is the efficiency of the Pol1B-HE in extending a primer with a DNA mismatch at the 3' end?

An interesting point – we believe that an appropriately thorough dissection of this issue is beyond the scope of the current paper, but we have raised the issue in the discussion – thanks for the suggestion.

Can the authors speculate/discuss whether this model holds true for the mammalian primase and the handoff to Pol alpha. A comparison of similarities would be welcome.

Given the lack of documented evidence for DNA synthesis by mammalian primase, and the documented preference for the eukaryotic innovation, Pol alpha, to extend RNA - DNA hybrids we suspect this mechanism may not apply to the eukaryotic system.

Some of the data/figures can be moved to the supplemental material. Fig 7 is a probability curve that should not be in the main text.

With respect, we would prefer to retain the probability curve as the model is integral to our proposed mechanism. We have, however, moved figure 3 to the supplemental material.

Reviewer #2 (Remarks to the Author):

The authors report that the DNA polymerase of *Sulfolobus solfataricus* PolB1-HE extends an RNA primer with significantly lower efficiency in comparison to a DNA primer, and the DNA-polymerizing activity of primase PriSLX significantly stimulates RNA primer extension by PolB1-HE. Based on a stochastic mode of the first dNMP insertion into an RNA primer, PriSLX affinity for ATP and dATP, and in vivo ratios of rNTPs and dNTPs, it was estimated that one dNMP should be incorporated into the 20-mer primer on average. The authors presented some evidence that after attachment of one dNMP to the 3'-end of an RNA primer, PriSLX switches into DNA polymerizing mode and generates a DNA tail, which promotes primer release by primase and its hand off to PolB1-HE. In Discussion section, it was speculated that the DNA tailing is the main mechanism of primer synthesis termination by PriSLX, while the previously proposed mechanism of RNA synthesis termination based on primer length (Yan et al., 2018 PNAS) serves as a backup.

The main and critical drawback of this work is comparison of RNA and DNA synthetic activities of PriSLX at physiologically non-relevant relative concentrations of rNTPs and dNTPs. For example, experiments presented in Figure 5 and intended to show the higher efficiency of PriSLX in DNA polymerizing mode versus the RNA polymerizing one were conducted at the same concentration of rNTPs and dNTPs (0.1mM). On the other hand, the authors mention that concentration of dNTPs comprises only 2.4% of the total nucleotide triphosphate pool in *Sulfolobus solfataricus*. Moreover, in the previous report of this group (Yan et al., 2018 PNAS), performed at physiologically relevant relative concentrations of nucleotide triphosphates, it was observed that PriSLX functions as an RNA synthetic enzyme. In particular, they wrote “did not find any compelling evidence for an ordered sequential synthesis of first RNA and then DNA by the primase”.

The assays the referee is referring to were conducted with primase alone and over 20-minute time frames – thus multiple rounds of distributive synthesis may have taken place. Even under those conditions we did not find evidence for long contiguous tracts of DNA synthesis – in agreement with our current work. Tracts of 2 or 3 sequential dNTPs would not have been readily resolved in the previous assays. We have emphasized this point in the revised manuscript

Actually, no evidence is provided that PriSLX switches into DNA polymerizing mode after attachment of one dNMP to the 3'-end of an RNA primer. In fact, Figure 5C shows that DNA polymerase activity of primase dramatically reduced on 18r/1d primer (lanes 29-31) in comparison to RNA primer (lanes 17-19). So, if to correctly interpret Figure 5C, it shows that addition of one dNMP to the RNA primer severely inhibits the following primer extension by PriSLX with ribo and deoxy NTPs. This is a very inefficient way to generate a DNA tail of 2-3 dNTPs.

We would argue that the inefficient nature of the extension is integral to the model that we are proposing – primase will readily disengage and thus render the 3' terminal dNMP's 3' OH group accessible to the replicative DNA polymerase. Note that even a single dNMP makes a primer a preferred substrate for elongation by PolB1-HE

The other claim that dNMP incorporation by PriSLX drives primer disengagement is not supported. There is no clear description what the term “disengagement” means and what value characterizes it, the dissociation constant (K_d) or the dissociation rate constant (K_{off}). The binding studies show similar K_d values for DNA, RNA and chimeric primers. In this case, K_{off} values for chimeric primers could be higher in comparison to RNA primer if dissociation rate constant (K_{on}) values are also increased but there is no clear reason for this. K_{on} is defined by diffusion rates for primase and substrate and depends on their size. If authors believe that DNA tailing increases K_{off} value, they should directly measure it for RNA, DNA and chimeric primers.

This is an important point – while the affinity for binding to a RNA or DNA tailed substrate is essentially unaltered, the dynamic behavior seen during synthesis (as revealed by our kinetic analyses) reveals a synthesis-coupled discrimination following NMP or dNMP incorporation. We infer that during translocation of the primer template junction in the enzyme's active site, a discriminatory step is taken resulting in a higher propensity to release a dNMP-tailed primer we have modified the text to address this point.

It was mentioned that primer disengagement somehow connected to the steady-state rate but no clear explanation and/or equation were provided.

We have included references to a number of papers that demonstrate that the steady state rate is dominated by primer release in a variety of polymerases.

In addition, if incorporation of just one dNMP into the RNA primer promotes primer disengagement, there is low probability that PriSLX would be able to incorporate a second and third dNMP in the presence of PolB1-HE. Thus, proposed disengagement mechanism contradicts with DNA tailing mechanism.

We have modified the text to emphasize that incorporation of even a single dNMP makes a primer a preferred substrate for elongation by PolB1-HE. If that +1 dNMP primer is not transferred to the DNA polymerase, subsequent additional dNMP incorporation by primase will provide a “second or third chance” for transfer. During nucleic acid chain elongation, substrate disengagement is not invariably set – but the propensity for the primase to release is much greater after dNMP than NMP incorporation as demonstrated in Figure 5b.

There are some issues with kinetic studies. Reactions with time points until 16 sec should be analyzed in the case of dGTP. It is important for the accuracy of burst assay that at reaction start an enzyme is saturated with substrates, which should be supplied at concentration ten times higher than corresponding K_d value. The RNA primer was supplied at concentration only 2.5 times higher versus the K_d value. A K_d value for PriSLX/GTP complex was not provided so it is not clear whether PriSLX was saturated with it. It is important that PriSLX affinity for substrates is measured at the same conditions (buffer, salt, temperature), which are used in kinetic assay. If doing these measurements is not feasible, it would be satisfactory if the

kinetic studies will be repeated at next concentrations: 10 μ M primer, 2.5 μ M PriSLX, 0.5mM (d)GTP.

Thank you for raising this point and giving us the opportunity to address it with these revised experiments.

The referee is absolutely correct – substrates should be supplied at a high excess over the K_d . When we first designed the RQF experiments we based the concentrations of the substrates on the K_d s that we had established using fluorescence anisotropy in the Yan et al., PNAS paper. Those results gave a K_d for nucleotide binding in the nanomolar range and for the primer template binding in the region of \sim 100 nM. Those fluorescence anisotropy assays were performed at 50 °C. For the current manuscript with the Cy5-labelled primer templates we did not have access to an appropriate filter set for fluorescence polarization measurements and so decided to perform microscale thermophoresis (MST). Regrettably, the temperature control system on the MST was broken (and remained so even after we sent it for repair) so we had to perform the MST measurements at room temperature – as the referee correctly emphasizes, far removed from the temperature at which the rapid quench flow experiments were performed. The K_d s that we determined using the MST were, as reported in our initial submission of the current manuscript, in the micromolar range.

Subsequently, we have gained access to a Synergy Neo2 instrument that has allowed us to perform the polarization measurements with the same substrates, the same buffers and at the same temperatures as we performed the rapid quench flow experiments. As can be seen in our revised manuscript – the fluorescence anisotropy data confirm our previous K_d measurements in the 100 nM range. We have removed the room temperature MST assays from the paper and replaced them with the new fluorescence anisotropy data. Importantly, these revised conclusions vindicate our initial experimental design for the RQF experiments, with substrate present at a concentration \sim 30-fold above the K_d .

Regarding the extension of the time-course to 16 seconds for the dGTP assay: by that time we see the effect of substrate depletion and so the data at these later time points are uninformative.

Interpretation of results presented on Figure 8 is misleading. Elongation of a 5'-triphosphate RNA-primer by PriSLX shows clear synthesis termination at 10-mer and not at 20-mer as authors concluded.

Our apologies for creating this confusion: at no point did we attempt to claim that primers were exclusively 20 nt in length. Rather, we have stated that primers are up to 20 nt in length. Unambiguously, primase is a limited processivity enzyme, but what can be clearly seen in Figure 6b is that primer elongation with dATP gives rise to a significantly shorter distribution of products than reactions with ATP. Crucially, the data in Figure 6c show that dATP exerts a dominant effect on constraining primer length even when present at a molar deficiency compared to ATP. We have altered the text to both clarify and emphasize these points.

The intensity of a 10-mer band is several fold higher versus the other bands (lanes 1 and 2). Moreover, the intensity of bands $>$ 10-mer is overestimated because the α -³²P-(d)ATP and the template with poly(dT) track were used in reaction resulting in higher specific radioactive activity for next extension product versus the previous one. For example, the

intensity of a 20-mer product is overestimated 6 times versus the 10-mer. For synthesis termination studies, it is important to label the primer only at one site in order to observe the true distribution of extension products. It is not clear why the authors decided do not use the same template as in previous study (Yan et al., 2018 PNAS), which has a poly(dA) track with one thymine and allows incorporation of only one α -³²P-(d)AMP into the primer. In addition, the level of products with length of >10 nucleotides would be further reduced in the presence of PolB1-HE, which will bind the 10-mer primer and prevent its extension by primase.

We note that the radio-label reactions in Figure 6b and c only contain ~6.5% radio-labeled nucleotides. Therefore, the average specific radioactivity of each subsequent extension product does not increase linearly as suggested above.

In Figure 6c, the assay does not investigate termination per se, but the ability of cold dATP titrant to out-compete a constant amount of radio-labeled α -³²P-ATP. Coherent with our model, we observe dATP out-competing the radio-label and a concomitant decrease in primer length.

Moreover, slow extension of RNA primer by PolB1-HE with the first dNMP might have a biological sense. Is it a unique feature of PolB1-HE or there are known examples of other DNA polymerases, which take the primer from primase and demonstrate the same discrimination against the RNA primer? Affinity of PolB1-HE for RNA, DNA and chimeric primers should be analyzed. If discrimination against the RNA primer is based on affinity, it might be not the issue in vivo where the local substrate concentration shall be high enough to saturate PolB1-HE.

We have performed the requested experiments using fluorescence polarization measurements for all four substrates. The results, presented as new Figure 2d, reveal similar binding affinities for PolB1-HE for all 4 substrates, with the strongest affinity for the RNA primer/template.

Reviewer #3 (Remarks to the Author):

The manuscript written by Greci MD et al is about biochemical analysis of the DNA replication mechanism in the hyperthermophilic archaeon, and described that PolB1-HE cannot utilize the 3'-RNA end to extend, but when PriSLX incorporates dNTP into the RNA end of the primer, PolB1-HE becomes to extend it. PriSLX easily dissociates from the 3'-DNA end, and therefore, a primer can be hand-off from PriSLX to PolB1-HE for further processive DNA synthesis during the DNA replication process. It is the first report describing the transferring mechanism from primase to replicase during the replication process in Archaea, and I think it is generally interesting for the readers of Nature Communication. I find several parts that I'm not completely satisfied with the descriptions by the authors. I hope the following comments will help to improve the manuscript before publication.

line 25. "vertebrates possess an AEP family member, PrimPol, that acts - - - "
I think PrimPol is also found from Archaea, bacteria, and virus. Revise this description precisely.

Thanks for catching this, we have revised the section accordingly.

line 64. "using purified recombinant Sulfolobus PriSLX and the replicative B-family DNA

polymerase holoenzyme, PolB1-HE.” I imagine from the surrounding description and citations that Sulfolobus seems to be *S. solfataricus*, but no description is found anywhere in the manuscript which organism was used in this study. Clarify it. In addition, *Sulfolobus solfataricus* has been recharacterized as ‘*Saccharolobus solfataricus*. It may be better to be updated.

Thanks for pointing this out and our apologies for the lack of clarity – we have added the species name to the introduction

line 118. PriSLX PD is first mentioned here, but it is already included in Figure 1. Please check the descriptions and figure numbers again in the whole manuscript.

Checked and corrected.

Figure 3a, lane 1-3, and 17-19 were the results from exactly same conditions as those of Figure 2c lane 1-3 and 4-6 (or Figure 2a, lane 17-19 and 20-22). However, they do not look the same. I think the consistency between the figures in the same manuscript is the essential issue for the good paper.

We thank the referee for raising the point. We have indeed noticed a small degree of variability (as noted by the referee, approximately 2-fold) in specific activity of different preps of the PolB1-HE. We emphasize that each experiment shown (which were all performed at least in triplicate) all contain internal controls. Importantly, the inclusion of internal controls in our assays allows us to comment on relative, rather than absolute activity levels.

The description for Figure 3b is missing in the text. The result of RNA primer-extension (Fig 3b) is clearly different from hybrid primer-extension (Fig 3a). BSA and PriSLX PD non-specifically promote the extension by PolB1-HE from the hybrid primer, but they do not look like promoting the RNA primer-extension. Clarify this point.

Related to the above comment, “very modest enhancement” in line 128 is really weak. I think the result in Figure 3 may be enough to be shown in a Suppl figure.

We agree and have moved this figure to Supplementary data.

Figure4, lane17–20. Why does the extension of 19 nt DNA, but not others, can be promoted by BSA? Some reasonable explanation is required.

There is in fact a modest stimulation of elongation of the 17r/2d and 18r/1d effected by inclusion of BSA.(Compare Supplementary Figure 2 lanes 1 and 4 and lanes 17 and 20).

line 170–174 “Remarkably, even though primers in our current assays possessed a 5'-florescent label, and thus are presumably not recognized by PriX, an overwhelming majority of 19nt RNA primer elongation remained at +3 nucleotides (22 nt product) for both RNA and DNA synthetic modes (Figure 5b – lanes 10 and 20), suggesting that structural aspects of length constraint remain engaged despite the fluorescent label.”

The authors' group published in their previous work that PriSLX containing PriX initiation site mutations, D70A, R72A or R74A, which lost the interaction with the 5'-end of the primer, lost the function of the caliper and the primer length was not limited in the Fig 3B of Yan et al., 2018, PNAS.

Respectfully, we disagree with this interpretation of our previous results. Examination of Figure 3B and Figure 4 in the Yan et al., PNAS paper reveals that when the caliper is active, primer yield is highest and length is most tightly constrained, however, even in the absence of the caliper, primers are typically of less than 25 nt in length – thus while the caliper clearly contributes to efficiency and aspects of length constraint, the primase is independently constrained in its ability to produce primers of greater than 25 nt.

However, in this study, they showed the data that PriSLX functions as a caliper without the interaction of PriX and 5'phosphate (Fig. 5B lane 8–10). Clarify this inconsistency and add some data to explain the result of Fig 5B.

For the reasons described above we do not believe there is any inconsistency with our previous data.

line 353. “Our model is also compatible with the recent report of the primase of *Pyrococcus abyssi* playing a role in lesion bypass¹⁵. When polymerizing from a 3'-DNA-terminus, the primase would be committed to DNA synthetic mode. Thus, acting as a DNA polymerase, the primase would be prone to disengage quickly. As primase lacks proof-reading activity, this would limit the length of the primase-synthesized DNA tract and thus minimize the likelihood of introducing mismatches during the repair process.”

I know that the research group published the paper of ref 15 also reported the results of a primer extension performed using *P. abyssi* PriSL and d/rNTP in a different paper. The synthesized product of dNTP alone is longer than that of rNTP alone or d/rNTP mixture in the report (*J. Mol Biol* 430, 4908-4924), which is the opposite of the results from *S. solfataricus*. Combined with other earlier studies of *P. furiosus* (*J Biol Chem* 276, 45484, 2001; *Nucleosides Nucleotides and Nucleic Acids*, 25, 681, 2006), I wonder if the presented model in this study could not be adapted to Euryarchaeota. More pondered discussion could be required if the conclusion is compatible to other archaea.

The experiments cited by the reviewer were performed for extended time periods and at temperatures ranging from 30 – 45 degrees below the optimal growth temperature for *Pyrococcus*. We note that in Figure 5B of the Liu et al 2001 JBC paper, no discernable DNA product is observed at a 3-minute-timepoint. Further, we note that in assays performed in the Lemor et al., 2018 JMB paper, enzyme is present at a 2-molar excess over substrate and incubation was performed 35 C below the growth temperature of *P. abyssi* for 30 minutes. Notably, the study only investigates extension of a DNA primer.. Due to the numerous and fundamental differences in experimental design, we find it hard to draw direct comparisons between the euryarchaeal studies and our own. Nevertheless, we have included discussion of the *Pyrococcus* data in our revised manuscript.

To confirm if the hypothesis proposed by the authors (Figure 9a–b) is true in *Sulfolobus*, I think the experiment using PriSLX initiation-site mutants (PriX D70A, R72A or R74A) should be performed and see if the caliper function does not actually work. I would like to ask the authors to do this experiment.

For the reasons described above we feel that these additional experiments are not required.

Fig 5C right side. Typo "+3 dMPs"

line 202. Typo "yield pf product"

line 244. Typo "it dies when"

Figure 8b. Explain "ATP*" in the legend.

Table S1. The 3'-end of 17r/2d should be rACG, not rArCG.

REVIEWERS' COMMENTS

Reviewer #1 (Remarks to the Author):

I am satisfied with the author's revision and comments. I have no further comments

Reviewer #2 (Remarks to the Author):

Response to critique was satisfactory.

Reviewer #3 (Remarks to the Author):

I have read the responses of the authors to the reviewers comments including mine to their original manuscript and read through their revised manuscript. I understand all the responses and found that the revised manuscript were well written. I now recommend it to be published in Nat Comm.